# Whole-Genome Sequencing Analysis Revealed High Genomic Variability, Recombination Events and Mobile Genetic Elements in *Streptococcus uberis* Strains Isolated from Bovine Mastitis in Colombian Dairy Herds

**DOI:** 10.3390/antibiotics14030297

**Published:** 2025-03-12

**Authors:** Paola A. Rios Agudelo, Julián Reyes Vélez, Martha Olivera Angel, Adam M. Blanchard, Yesid Cuesta Astroz, Arley Caraballo Guzmán, Giovanny Torres Lindarte

**Affiliations:** 1Instituto Colombiano de Medicina Tropical, Universidad CES, Cra 43a No. 52 Sur 99, Sabaneta 055450, Antioquia, Colombia; prios@ces.edu.co (P.A.R.A.); jreyesv@ces.edu.co (J.R.V.); ycuesta@ces.edu.co (Y.C.A.); acaraballo@ces.edu.co (A.C.G.); 2Biogenesis Research Group, Faculty of Agriculture Science, University of Antioquia, Cra 75 No. 65-87, Medellín 050010, Antioquia, Colombia; martha.olivera@udea.edu.co; 3School of Veterinary and Science, University of Nottingham, Sutton Bonington Campus, Leicestireshire LE12 5RD, UK; adam.blanchard@nottingham.ac.uk; 4Escuela de Microbiología, Universidad de Antioquia, Ciudad Universitaria, Calle 67 No 12 53-108, Medellín 050010, Antioquia, Colombia

**Keywords:** bovine mastitis, *Streptococcus uberis*, whole genome sequencing, recombination events, mobile genetic elements

## Abstract

**Introduction:** *Streptococcus uberis* is a poorly controlled cause of bovine intramammary infections and a common motivation for the use antibiotics in dairy farms worldwide. Therefore, studying the genomic characteristics of this pathogen is fundamental to understand its complex epidemiology and behavior against antimicrobials. **Methods:** A comparative genomic analysis of 10 *S. uberis* strains was performed and their antimicrobial susceptibility was assessed. **Results:** Ten different novel sequence types were found, and genes (*tetM, tetO*, *patB*, *lnuC*, *lnuA*, *lsaE*, *ermB*, *ANT*(6)-la) and mobile genetic elements previously associated with antimicrobial resistance (repUS43, IS*Sag*2, and IS*Enfa*4) and virulence (315.2 phage) were detected. Additionally, our strains had the highest relative rate of recombination to mutation (8.3) compared to other *S. uberis* strains isolated from different continents (America: 7.7, Asia: 2.9, Europe: 5.4, and Oceania: 6.6). Most of the strains (80%) tested showed phenotypic resistance to clindamycin and 70% exhibited intermediate susceptibility to penicillin. **Conclusions:** The high heterogeneity of strains observed and the presence of genetic factors linked to antimicrobial resistance represent a challenge for the implementation and surveillance of measures focused on the control and elimination of this pathogen.

## 1. Introduction

*Streptococcus uberis* is the most common mastitis pathogen in many countries with a trend that appears to be increasing globally [1,2,3]. This bacterium is a commensal which can cause disease [4,5,6]. These behaviors have been linked to the capacity of *S. uberis* to adapt and survive in different environments, promoted by a great metabolic flexibility, revealing an evolutionary dynamic [7,8]. In addition, *S. uberis* can cause a persistent intramammary infections (IMI) due to its ability to evade the host immune response and the antimicrobials actions [9,10,11]. The proportion of antimicrobial resistance in *S. uberis* varies widely worldwide, which can be disseminated by mobile genetic elements such as plasmids, transposons, insertion sequences or phages [11]. These issues make *S. uberis* challenging to control in herds, impacting the sustainability of the modern dairy industry [6].

Genomic typing of *S. uberis* has allowed the identification of genetic profiles to highlight the relationship to clinical outcomes, with virulence factors and mobile genetic elements in strains isolated from IMI [7,12,13]. Those findings have revealed a huge genetic variability among herds and countries [14].

Genotyping tools used to study *S. uberis* include techniques such as pulsed field gel electrophoresis (PFGE), dot blot hybridization, multi-locus sequence typing (MLST) and, most recently, whole genome sequencing (WGS) [15,16,17,18]. However, WGS has advantages over the other named techniques because it is possible to compare the entirety of the *S. uberis* genome, making it a powerful analysis tool. This has become an insight strategy to identify patterns, genetic variations and monitoring [18]. Nevertheless, not all regions of the world have reported the genomes of *S. uberis* strains that have caused IMI. Therefore, the aim was to study the genetic diversity genetics and antimicrobial resistance profiles of *S. uberis* strains isolated from bovine mastitis in Colombian dairy herds by WGS and phenotypic testing.

## 2. Results

### 2.1. Streptococcus uberis Strains Isolated and Sequenced

A total of 21 (14%) *S. uberis* strains were isolated from the 150 samples evaluated and confirmed by PCR. In total, 10 of the 21 strains were selected, and their genomes sequenced. Of the strains sequenced, 30% (3/10) belonged to San Pedro de los Milagros, 20% (2/10) to Santa Rosa de Osos, 20% (2/10) to Entrerríos, 20% (2/10) to La Ceja, and 10% (1/10) to San Jerónimo. Isolates m2 and m3 were recovered from herd 1, m5 and m11 from herd 2, m6 and m9 from herd 3, m6 and m7 from herd 4, and m10 and m12 from herds 5 and 6, respectively. The draft genomes were deposited in BV-BRC and PubMLST (Table 1).

### 2.2. General Genome Features

Table 2 shows the genomic features of the 10 *S. uberis* strains sequenced and assembled. The genomes’ lengths ranged from 1,902,784 to 2,134,443, and the GC contents ranged from 36.72% to 37.02%. Strain m7 exhibited the highest nucleotide identity (98.81%), while strain m6 exhibited the lowest nucleotide identity (98.40%) compared to the pathogenic strain 0140J (GenBank accession: AM946015.1).

### 2.3. Pangenome Analysis

Analysis of the 10 isolates revealed a total of 2889 genes, with the core genome containing 1608 genes (55.65% present in the 10 strains). The shell genome consisted of 543 genes (18.79% present between 2 and 9 strains), and a cloud genome of 738 genes (25.54%) (Table 3). Specific genes (cloud genes) were found in each of the 10 strains evaluated, with strain m3 being the one that was most present (188/738), while strain m10 was the one that was least present (14/738) (Appendix A).

### 2.4. Multi-Locus Sequence Typing (MLST)

Ten different STs were assigned to each *S. uberis* strain included. All STs were considered novel, as they did not match any known ST previously reported in the *S. uberis* MLST database (https://pubmlst.org/organisms/streptococcus-uberis, accessed on 11 November 2024). Clonal complex (CC) 143 was only assigned to isolates m7 and m10 (Table 4).

The Phyloviz tree (Figure 1) shows the STs identified in this study, which were distributed and clustered in different branches of the tree. Despite the high level of diversity observed in our study, most strains (70%) clustered in the same branch, which also included *S. uberis* strains isolated from Australia, the United Kingdom (UK), the USA, China, and Denmark. Notably, isolate m3 was the only one that did not cluster with any other of the nine isolates evaluated. This strain clustered with another originating from Australia and the UK. Particularly, m3 was the isolate that showed the most specific genes (188/738) and the second one in which the most recombinant sequences (156) were found (Appendix A).

### 2.5. Recombination Events

The mean fragment size of the recombination events of our strains was larger than that of the other continents (753 bp) (Table 5). The recombination coverage was estimated to be 0.14 in our strains. This value means that 14% of the sites in the core genome sequence of each strain originated from recombination events. The recombination ratio in our strains was (8.28), which is higher than that of strains from other continents. This means that in our strains the recombination events were 8.28 times more frequent than point mutations.

### 2.6. Virulence Factors

Thirty-nine virulence genes were identified using the PATRIC platform (now known as BV-BRC). Isolates m2, m3, m8, and m9 contained 37 genes associated with virulence, the isolates m5, m6, m7, m10, m11 contained 38 genes associated with virulence, only isolate m12 presented 39 genes, with a single different gene (SP_0320). Some of the genes found and that have previously been reported in *S. uberis* were *purN, purH, purB, lepA, perR, leuS, vicK, sodA, cpsY, pauA, sua, lbp, gapC, and hasC* (Appendix A).

### 2.7. Mobile Genetic Elements (MGE)

Transposons were not detected in any of the 10 isolates. Analysis of the phages revealed that only two isolates carried intact prophages. Isolate m3 contained the phage YMC and m8 carried phage 315.2. None of the phages found have been associated with antimicrobial resistance. On the contrary, phage 315.2 has been linked with virulence, since has been detected in the serotype M3 of *Streptococcus pyogenes*, which is a common cause of severe invasive infections with unusually high rates of morbidity and mortality [19]. The PlasmidFinder detected the replicon plasmid sequence repUS43 in isolate m2. In the remaining nine strains, these MGE were not found.

Regarding insertion sequences (IS), four isolates (m2, m6, m11, and m12) did not contain these elements. However, IS*Sag*2 was detected in isolates m3, m5, m7, m8, m9, and m10. In addition, isolate m7 presented IS*Stin*10 and isolate m8 carried two ISs, IS*Efm2* and IS*Enfa*4 (Table 6).

### 2.8. Resistance Genes

A total of nine resistance genes (*tetM, tetO*, *patB*, *lnuC*, *lnuA*, *IsaE*, *ermB*, and *ANT*(6)-la) were found in the ten isolates. These genes confer resistance to tetracyclines (*tetM, tetO*), fluoroquinolones (*patB*), lincosamides (*lnuC*, *lnuA, lsaE*), macrolides (*ermB)* and aminoglycosides (*ANT*(6)-la). All isolates carried at least two resistance genes. The *lnu*C gene was found in 80% (8/10) and the *lnu*A gene in 10% (1/10) of the genomes, conferring resistance to antibiotics belonging to the lincosamide family. This finding is consistent with what was observed phenotypically, since most isolates were resistant to clindamycin, except isolates m8 and m9. Isolate m11 was the only one in which none of the genes that confer resistance to lincosamides were detected (Table 7).

The *tetM* and *tetO* genes were present in the genomes of isolates m2 and m6, respectively; both isolates showed phenotypic resistance to tetracycline. Isolate m6 also carried the *ErmB* gene, which confers resistance to erythromycin, as observed in the AST.

### 2.9. Antimicrobial Susceptibility Testing (AST)

All isolates tested were susceptible to ampicillin, cefotaxime, ceftriaxone, chloramphenicol, linezolid, and levofloxacin. On the contrary, 80% of the isolates showed resistance to clindamycin, except isolates m8 and m9, which were susceptible. Isolates m6 and m12 were resistant to erythromycin. Isolates m2 and m6 were resistant to tetracycline. For the remaining isolates, 50% exhibited intermediate susceptibility to erythromycin and only isolates m8, m9, and m10 were susceptible. For tetracycline, 80% of the remaining isolates were susceptible. For penicillin, 70% presented intermediate susceptibility (Table 8).

## 3. Discussion

In Colombia, *S. uberis* has become one of the most commonly identified pathogens responsible for bovine mastitis, which is poorly controlled [6,18]. The advanced ability of this pathogen to adapt to different environmental conditions and bovine body sites has been associated with its wide genetic diversity [18,20], which could behave as opportunists capable of causing intramammary infections in cattle [6]. Our finding evidenced the heterogeneity of circulating *S. uberis* strains in the region because all the S*. uberis* strains sequenced were different and classified as novel genotypes (STs not reported previously). Recently, a study confirmed the high diversity of this bacterium worldwide. In total, 1037 whole genomes of *S. uberis* isolated from 11 countries were analyzed between 1970 and 2022 using a core genome multi-locus sequence typing (cgMLST) scheme [6]. The results showed 932 cgSTs, double what was found with MLST (458 STs). The marked difference was attributed to high-resolution provided by cgMLST, since, with this approach, 1447 genes were compared, which is a significantly larger number of loci than used in MLST [6].

Our findings indicated that the *S. uberis* strains evaluated carried a common set of genes that encode different virulence factors responsible for the bacterial invasion in the mammary gland and promote its survival inside the host [21,22]. However, this is not sufficient evidence to suggest the pathogenicity level of the isolates tested or infection source. Several studies have reported the existence of strains adapted to the mammary gland and other environmental strains that can also cause intramammary infections [23,24], suggesting that the strains’ diversity in a herd is indicative of its mode of transmission [25]. Currently, these attributions are questionable and can mislead farmers and veterinarians during the implementation of effective anti-mastitis measures [25]. In the same study carried out with 1037 whole genomes of *S. uberis*, the results supported the concept that infection would be basically opportunistic and not due to transmission of specific clones from cow to cow or an environmental source [6].

Genetic recombination events contribute to diversification of distinct lineages in each *Streptococcus* species [26]. It also facilities the successful adaptation in bacteria inhabiting a wide range of niches and which are exposed to pressures from host (immune response or antibiotic treatment) or environmental changes, as is the case of *S. uberis* [8,27]. Although information on recombination and mutation rate is available for other *Streptococcus* species (*S. agalactiae*, *Streptococcus suis*, *S. pyogenes*, and *Streptococcus pneumoniae*) worldwide, it remains unknown for *S. uberis* isolated from bovine mastitis in Colombia. The estimation revealed that the recombination events occurred 8.28× more than the point mutations in our strains. This ratio was higher than *S. uberis* genomes from other continents compared in this study. Likewise, this ratio was higher than the value previously calculated for *S. pyogenes*, *S. suis*, and *S. pneumoniae* [26]. However, when the genome fraction derived from recombination events was determined; the result was 14.0%, lower than other *S. uberis* genomes compared (between 18 and 27%) and 44.5%, 17.2%, and 33.0% exhibited by *S. agalactiae*, *S. pyogenes*, and *S. suis*, respectively [26]. This was only higher than that calculated for *S. pneumoniae* (9.0%). In addition, the mean fragment size of recombination events was 753 bp, which is longer than reported for *S. pyogenes* (389 bp), *S. pneumoniae* (560 bp), and other *S. uberis* genomes compared (between 445 bp and 637 bp), but shorter than that found in *S. agalactiae* (1303 bp) and *S. suis* (3147 bp) [26].

Some of the mobile genetic elements (MGE) that play an important role in the recombination process are insertion sequences (ISs) and phages, which could be acquired from other bacterial species or genera [26]. Several ISs were found in most of the isolates sequenced, especially in strain m8, since this carried three different ISs. IS*Sag*2 was present in 60% (6/10) of the isolates and had its origin in *S. agalactiae* (https://isfinder.biotoul.fr/scripts/ficheIS.php?ident=2081, accessed on 17 October 2024). Interestingly, this element flanks a transposon harboring the *lmb* gene, implicated in invasive human infection [28]. Usually, the *lmb* gene is present in *S. agalactiae* recovered from human, but not in isolates from animals. Recently, we showed that *lmb*-carrier *S. agalactiae* isolated from bovines did not adequately respond to the antimicrobials administered to treat mastitis, likely because this virulence factor favored the cell invasion of these strains [29]. Another IS found was IS*Enfa*4 (https://isfinder.biotoul.fr/scripts/ficheIS.php?name=ISEnfa4, accessed on 17 October 2024), but unlike the previous one, this has been linked to plasmid pW9-2 from *Enterococcus faecalis* derived from sewage at a pig farm. it has also been associated with the *cfr* gene, which confers resistance to five antimicrobial classes (phenicols, lincosamides, oxazolidionones, pleuromutilins, and streptogramin A). It is important to note that our study did not find transposons, plasmids, or genes (*cfr* and *lmb*) associated with these ISs (IS*Sag*2 and IS*Enfa*4). Nevertheless, these findings suggest the mobilization of the genetic elements among bacterial species.

Regarding phages, intact prophages were detected in only two isolates, specifically YMC phage in strain m3 and phage 315.2 in strain m8. The YMC phage was initially reported in *Streptococcus salivarius*, a natural inhabitant of the human mouth [30], while phage 315.2 was characterized from the serotype m3 of the *S. pyogenes*, a strain related to severe invasive infections in humans [19].

The cure rate of the *S. uberis* infections can vary between 64% and 91%, since the effectiveness of treatments depends on cow characteristics, such as parity and multiple high SCC records, or strain properties like antibiotic resistance genes [11,31]. In this study, a limited number of antibiotic resistance genes were detected, *lnuC* being the most frequent. This gene encodes a transposon-mediated nucleotidyltransferase, which is responsible for the resistance to lincosamide antibiotics, which was initially found in *S. agalactiae*, especially in those strains isolated from humans. This suggests a recent interspecies jump by a lateral gene transfer [32]. The co-occurrence of the strains’ resistance to both lincosamides and macrolides such as erythromycin, which is mediated by *erm* genes, has also been described [33,34,35]. In our study, only two isolates showed resistance to these antibiotics; however, both isolates presented genes (*lnuC***^+^** and *ermB***^+^**) involved in these resistances. Although it is important to note that five strains clindamycin resistant (*lnuC***^+^** and *ermB***^−^**) exhibited intermediate susceptibility to erythromycin. Contrary to our finding, tetracycline resistance has been the most reported in other studies [11,36]. Tetracycline is widely used in livestock and their slow degradation can increase selective pressure for this antimicrobial [37]. Some of the genes (*tetM* and *tetO*) that can cause resistance to tetracycline were only found in two isolates. Notably, the finding of the replicon plasmid sequence repUS43 in isolate m2 aligns with the presence of the *tetM* gene and tetracycline resistance observed in this strain. It has previously been reported that repUS43 can harbor antibiotic resistance genes such as *vanA*, *tetM*, and *ermB*, which are commonly found in *Enterococcus* spp. isolated from different sources (environment, livestock, and clinical) [38,39,40]. These reports suggest that the repUS43 play an important role in tetracycline resistance dissemination, even among different bacterial species, as was observed in our study [39,41]. Moreover, tetracycline resistance has also been observed in *S. agalactiae* clones adapted to humans and cattle [42], so DNA exchange could occur in their common niche [42]. Most (70%) isolates showed intermediate susceptibility to penicillin, but all (100%) were susceptible to ampicillin. Antimicrobial efficacy in those isolates with intermediate susceptibility will likely be reduced in vivo or evolve into resistant phenotypes [34].

## 4. Methods and Materials

### 4.1. Sample Collection

From November 2020 to June 2021, a total of 150 samples of milk were obtained from dairy cows belonging to six specialized dairy herds located in the department of Antioquia, Colombia (Table 9).

Milk samples were collected from cows with mastitis during the lactation period and were not receiving antibiotic treatment. These samples were collected using a sterile container according to recommendations from the National Mastitis Council (NMC) [43] which were sent to the Industrial Microbiology Laboratory of the Colombian Institute of Tropical Medicine (ICMT)-CES University for isolation and identification of *S. uberis*.

### 4.2. Streptococcus uberis Isolation

Ten microliters of milk was plated on esculin blood agar (EBA) (Scharlab, Sentmenat, Spain) and enterococci agar (Becton Dickinson, Franklin Lakes, NJ, USA). The plates were then incubated aerobically at 37 °C for 24–48 h. Bacteriological cultures were interpreted based on standard protocol recommended by NMC [43]. A presumptive identification of *S. uberis* was considered if the isolates were gram-positive cocci, catalase negative, esculin was hydrolyzed on EBA, and no growth on enterococci agar was observed. The presumptive *S. uberis* strains were stored in trypticase soy broth (Scharlab, Sentmenat, Spain) with glycerol at –80 °C.

### 4.3. DNA Extraction and S. uberis Identification by PCR

Total DNA was extracted from presumptive *S. uberis* strains using the GeneJET Genomic DNA Purification Kit (Thermo Fisher Scientific™, Waltham, MA, USA) according to the manufacturer’s instructions. The concentration and purity of the extracted DNA was determined by NanoDrop 2000 (Thermo Fisher Scientific™, Waltham, MA, USA). Subsequently, the *S. uberis* strains were confirmed by Polymerase Chain Reaction (PCR) using the primers described by Hassan et al. (2001), which allowed amplification of a fragment of the 16S rRNA gene: F-CGCATGACAATAGGGTACA/R-GCCTTTAACTTCAGACTTATCA [44].

PCR was performed using a Master Mix One Taq^®^ 2× MM w/Standard Buffer M0482S (New England Biolabs, Ipswich, MA, USA) and a final reaction volume of 25 µL. The reaction mixture (25 µL) contained 12.5 µL of Master Mix, 0.5 µL of each primer, 6.5 µL of distilled water, and, finally, 5 µL of DNA. The parameters used for amplification were 94 °C for 60 s, 94 °C for 60 s, 58 °C for 90 s, 72 °C for 90 s, and a final extension of 72 °C for 5 min, for 30 cycles [25]. The presence of PCR products was confirmed by agarose electrophoresis gel prepared at 1% and run on Tris Acetate Electrophoresis Tampon (TAE). The reference strain of *S. uberis* ATCC 700407 was used as positive control. DNA was stored at −80 °C until DNA sequencing.

### 4.4. Whole Genome Sequencing, Assembly and Annotation

Genomic DNA Sequencing and standard data analysis (Quality control of raw sequencing data and mapping to reference genome) were performed by Novogene (Sacramento, CA, USA). Pair reads of 150 bp in length were obtained after library preparation using NEBNext^®^ Ultra^TM^ II DNA Library Prep Kit (New England Biolabs, Ipswich, MA, USA) and then sequenced using the Illumina NovaSeq X Plus (Illumina, San Diego, CA, USA).

De novo assembly was performed with SPAdes (version 3.15.5). To correct assembly errors, the genomes were annotated using RAST (Rapid Annotations using Subsytems Technology —version 1.073), a tool available in PATRIC 3.6.12 (https://patricbrc.org/, accessed on 10 October 2024) [45].

### 4.5. Genome Analysis

Pangenome analysis was performed using Roary (version 3.11.2) [46]. The genomes were searched for virulence factors using the PATRIC 3.6.12 service for specialized genes and filtered by the “virulence factor” option [45]. In addition, the resistance genes of the genomes were searched in the Comprehensive Antibiotic Resistance Database (CARD) (https://card.mcmaster.ca/, accessed on 18 November 2024) [47].

The PlasmidFinder 2.1 tool was used for plasmid detection (https://cge.food.dtu.dk/services/PlasmidFinder/, accessed on 18 November 2024) [48]. The PHASTER web server (https://phaster.ca/, accessed on 18 November 2024) [49] was used to identify bacteriophages. The insertion sequences and transposons were identified using the Mobile Element Finder tool v1.0.3 (https://cge.food.dtu.dk/services/MobileElementFinder/, accessed on 18 November 2024) [50]. The mcorr program was used to detect recombination events [51].

The average nucleotide identity (ANI) was calculated by JSpeciesWS (http://jspecies.ribohost.com/jspeciesws/#anib, accessed on 17 October 2024) [52], using the *S. uberis* strain 0140J as reference.

### 4.6. Recombination Events Detection

The recombination rates of the core genomes were estimated using the mcorr program (version 20180314) with default parameters [26]. The core genome alignment of each *S. uberis* strain was used as input with 1000 bootstrapped replicates [26]. Our isolates were compared with 40 different *S. uberis* strains, 10 from each continent (North America, Asia, Europe, and Oceania). The estimated parameters were the mutational divergence (θ), recombinational divergence (φ), recombination coverage (c), mean recombination fragment size (f), diversity (d), and relative rate of recombination to mutation (φ/θ) [26].

### 4.7. Multi-Locus Sequence Typing (MLST)

Genomes assembled and stored in FASTA format were submitted to PubMLST (https://pubmlst.org/organisms/streptococcus-uberis, accessed on 11 November 2024) in order to determine the sequence types (STs) and clonal complexes (CCs). Additionally, for a more comprehensive analysis of the possible patterns of evolutionary descent, an eBURst analysis was performed using STs obtained in this study and other STs previously published in the *S. uberis* PubMLST database (https://pubmlst.org/organisms/streptococcus-uberis, accessed on 11 November 2024).

### 4.8. Antibiotic Susceptible Testing

The minimum inhibitory concentrations (MICs) of antimicrobial agents were determined using AST-GP67 card (*Streptococcus*-type Gram-positive bacteria) of the VITEK^®^ 2 system (Biomerieux, Marcy-l’Étoile, France). The 10 antibiotics tested were ampicillin, penicillin, cefotaxime, ceftriaxone, chloramphenicol, clindamycin, erythromycin, levofloxacin, linezolid, and tetracycline. The results were interpreted according to the Clinical Laboratory Standards Institute’s (CLSI) guidelines for veterinary antimicrobial susceptibility testing [53]. *Streptococcus pneumoniae* ATCC 49619 was used as quality control strain.

### 4.9. Statistical Analysis

Descriptive analyses were performed using R statistical software (version 4.4.0) to assess the genotypic and phenotypic characteristics of antimicrobial susceptibility, genetic variability, and other genomic attributes. Due to the low number of isolates, relevant statistical comparisons were not possible.

## 5. Conclusions

The high genetic diversity and the rate of recombinant events found in these *S. uberis* strains highlight their ability to acquire genetic material that could influence their pathogenicity and antimicrobial resistance, as well as the evolutionary and adaptive dynamics of the bacteria. This underlines the need to adjust infection control strategies in dairy farms, using a comprehensive strategy that combines genomic surveillance, antimicrobial resistance control, and new therapeutic strategies.

## 6. Limitations

A limitation of this study was the limited number of isolates sequenced.

## Figures and Tables

**Figure 1 antibiotics-14-00297-f001:**
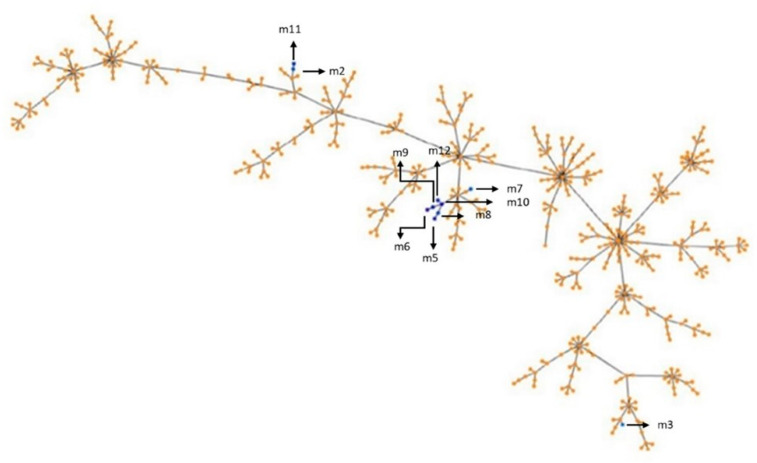
Phyloviz tree based on STs. The blue dots and arrows represent STs assigned in this study. The orange dots represent all other *S. uberis* STs published in the PubMLST database.

**Table 1 antibiotics-14-00297-t001:** Distribution of the *S. uberis* isolates sequenced by municipality and herd.

Isolate	Municipality	Herd	BV-BRC/PubMLST ID
m2	La Ceja	1	218495.87/2323
m3	La Ceja	1	218495.88/2324
m5	San Pedro de los Milagros	2	218495.92/2325
m6	Entrerríos	3	218495.84/2326
m7	Santa Rosa de Osos	4	218495.83/2327
m8	Santa Rosa de Osos	4	218495.85/2328
m9	Entrerríos	3	218495.93/2329
m10	San Jerónimo	5	218495.94/2330
m11	San Pedro de los Milagros	2	218495.95/2331
m12	San Pedro de los Milagros	6	218495.96/2332

ID: identification.

**Table 2 antibiotics-14-00297-t002:** Genomic features of the *S. uberis* strains isolated from bovine mastitis.

Isolate	Genome Size (bp)	CDS	GC Content (%)	Contigs	N50	ANI (%)	Antibiotic Resistance Genes	Virulence Factors	Plasmids	Transposons	Insertion Sequences	Prophages
m2	1,953,684	1959	36.77	35	446,257	98.68	3	37	1 *	0	0	0
m3	2,036,297	2038	36.94	26	446,687	98.49	2	37	0	0	1	1
m5	1,952,255	1957	37.02	42	428,101	98.70	2	38	0	0	1	0
m6	2,134,443	2161	36.95	102	415,269	98.40	6	38	0	0	0	0
m7	1,964,012	1956	36.77	36	412,612	98.81	2	38	0	0	2	0
m8	2,053,252	2096	36.88	77	1,076,464	98.60	3	37	0	0	3	1
m9	1,967,434	1953	36.75	45	420,690	98.80	2	37	0	0	1	0
m10	1,902,784	1895	36.88	38	375,227	98.72	2	38	0	0	1	0
m11	1,956,456	1944	36.88	38	434,788	98.69	3	38	0	0	0	0
m12	1,947,552	1933	36.72	40	428,747	98.72	3	39	0	0	0	0

bp: base pairs; CDS: coding sequences; GC: guanine and cytosine content; ANI: average nucleotide identity; * Replicon plasmid sequence repUS43.

**Table 3 antibiotics-14-00297-t003:** Pangenome analysis results.

Pangenome	Gene Count	Percentage (%)
Core genes	1608	55.65
Soft core genes	0	0
Shell genes	543	18.79
Cloud genes	738	25.54
Total	2889	100

**Table 4 antibiotics-14-00297-t004:** MLST allelic profiles and STs assigned.

MLST Allelic Profiles										
Isolate	ST	Genes							CC	Herd
		*arcC*	*ddl*	*gki*	*recP*	*tdk*	*tpi*	*yqil*		
m2	1426	6	1	3	2	42	52	3	NA	1
m3	1427	6	1	5	2	17	52	6	NA	1
m5	1428	1	1	3	1	42	52	6	NA	2
m6	1435	1	1	2	2	28	52	86	NA	3
m7	1429	6	1	28	2	17	3	3	143	4
m8	1430	1	1	3	2	130	52	6	NA	4
m9	1431	1	1	28	2	28	52	3	NA	3
m10	1432	1	1	3	2	17	52	3	143	5
m11	1433	6	1	3	2	131	52	86	NA	2
m12	1434	1	1	3	2	26	52	3	NA	6

ST: sequence type assigned; CC: clonal complex assigned; NA: Not assigned.

**Table 5 antibiotics-14-00297-t005:** Recombination parameters estimated by mcorr.

Parameter	Colombia	America	Asia	Europe	Oceania
θ	0.0202	0.0140	0.0207	0.0161	0.0176
φ	0.1676	0.1093	0.0602	0.0872	0.1182
f	753	567	445	632	637
c	0.1424	0.2111	0.2102	0.1789	0.2679
d	0.0028	0.0029	0.0044	0.0029	0.0046
φ/θ	8.2814	7.7696	2.9065	5.4080	6.6949

**Table 6 antibiotics-14-00297-t006:** Insertion sequences detected in the isolates evaluated.

Insertion Sequence	Isolate									
	m2	m3	m5	m6	m7	m8	m9	m10	m11	m12
IS*Sag*2		X	X		X	X	X	X		
IS*Stin*10					X					
IS*Efm*2						X				
IS*Enfa*4						X				

X: presence of insertion sequence.

**Table 7 antibiotics-14-00297-t007:** Genes associated with antimicrobial resistance found in the isolates evaluated.

Gene	Isolate									
	m2	m3	m5	m6	m7	m8	m9	m10	m11	m12
*tetM*	X									
*tetO*				X						
*patB*	X	X	X	X	X	X	X	X	X	X
*lnuC*	X	X	X	X	X	X	X	X		
*lnuA*										X
*lsaE*				X					X	
*ANT*(6)-la				X					X	
*ermB*				X						X

X: presence of the gene.

**Table 8 antibiotics-14-00297-t008:** Results of the antimicrobial susceptibility test by isolates and antibiotics.

Antibiotic	Isolate									
	m2	m3	m5	m6	m7	m8	m9	m10	m11	m12
Ampicilin	S	S	S	S	S	S	S	S	S	S
Penicilin	S	S	I	I	S	I	I	I	I	I
Cefotaxime	S	S	S	S	S	S	S	S	S	S
Ceftriaxone	S	S	S	S	S	S	S	S	S	S
Levofloxacin	S	S	S	S	S	S	S	S	S	S
Chloramphenicol	S	S	S	S	S	S	S	S	S	S
Linezolid	S	S	S	S	S	S	S	S	S	S
Clindamycin	R	R	R	R	R	S	S	R	R	R
Erythromycin	I	I	I	R	I	S	S	S	I	R
Tetracycline	R	S	S	R	S	S	S	S	S	S

S: susceptible; I: intermediate susceptible; R: resistant.

**Table 9 antibiotics-14-00297-t009:** Samples distribution.

Herd	Municipalities	Number of Samples
1	La Ceja	30
2	San Pedro de los Milagros	40
3	Entrerríos	12
4	Santa Rosa de Osos	16
5	San Jerónimo	2
6	San Pedro de los Milagros	50
	Total	150

## Data Availability

The *S. uberis* genomes are available in the BV-BRC (https://www.bv-brc.org/, accessed on 10 October 2024) under IDs: 218495.87, 218495.88, 218495.89, 218495.90, 218495.91, 218495.92, 218495.93, 218495.94, 218495.95, 218495.96, and PubMLST database (https://pubmlst.org/, accessed on 11 November 2024) under IDs: 2323, 2324, 2325, 2326, 2327, 2328, 2329, 2330, 2331, 2332.

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
