# Peer review of "Whole-Genome Sequencing Analysis Revealed High Genomic Variability, Recombination Events and Mobile Genetic Elements in *Streptococcus uberis* Strains Isolated from Bovine Mastitis in Colombian Dairy Herds"

_antibiotics, 2025, doi:10.3390/antibiotics14030297_

Round 1

Reviewer 1 Report

Comments and Suggestions for Authors

The study by Paola A. Ríos Agudelo  et al has reported the genetic diversity  and antimicrobial resistance profiles  in S.  uberis strains isolated from bovine mastitis in Colombian dairy herds. The study  provides crucial  genomic data in S.  uberis associated with bovine mastitis, and that can provide basic information to deepen the understanding of the complex epidemiology and behavior of S.  uberis against antimicrobials.  The manuscript is well-written and interesting. However, there are some issues and limitations we spotted in the manuscript that need to be addressed before considering this manuscript for publication.

 Comments to Authors

1)      The study focuses on only 10 strains of S. uberis, which may not be representative of the entire diversity of the pathogen. Therefore, I suggest analyzing and characterized all isolated (21) to provide a clear and wider picture of the pathogen diversity

2)      The present study also needs to investigate and characterize another element behind the phenotypic resistance such as porin mutation and efflux pump.

3)      The conclusion part is poorly discussed and does not provide comprehensive information on the entire study and findings.

4)      Please provide more information on statistical analysis

5)      There is no mentioned of a quality control test used in the study when performing AST test 

Author Response

Thank you for your comments and suggestions.

Comments 1: I suggest analyzing and characterized all isolated (21) to provide a clear and wider picture of the pathogen diversity.

Response 1: We understand this requirement; however, at this moment it is not possible to carry out additional assays to characterize all isolated, because we currently don't have the financial resources to perform these analysis. Nevertheless, it is important to highlight that the strains were randomly selected according to the number of isolates obtained per farm. Additionally, high genetic diversity was observed using around 50% of the sequenced strains, so probably if we had sequenced all strains, similar results would have been obtained, as demonstrated in one of studies cited in our manuscript. The authors analyzed 1,037 S. uberis genomes, finding 932 cgSTs.

Comments 2: The present study also needs to investigate and characterize another element behind the phenotypic resistance such as porin mutation and efflux pump.

Response 2: The search in specialized database for genetic factors involved in antimicrobial resistance included all known mechanisms (e.g., efflux pump, mutations, antibiotic inactivation enzyme). Most genes found confer resistance through antibiotic inactivation enzymes and antibiotic target protection proteins. The patB gene was the only one that presents a different mechanism, since it confers resistance through an efflux pump mechanism.

Comments 3: The conclusion part is poorly discussed and does not provide comprehensive information on the entire study and findings.

Response 3: The conclusion was adjusted (lines 345 - 350).

Comments 4: Please provide more information on statistical analysis.

Response 4: The statistical analysis was adjusted (lines 340 - 343).

Comments 5: There is no mentioned of a quality control test used in the study when performing AST test.

 Response 5: The bacterial strain (S. pneumoniae ATCC 49619) used in AST quality control was included in the manuscript (lines 338 - 339).

Reviewer 2 Report

Comments and Suggestions for Authors

The manuscript titled "Whole-genome sequencing analysis revealed high genomic variability, recombination events and mobile genetic elements in Streptococcus uberis strains isolated from bovine mastitis in Colombian dairy herds" focuses on comparative genomic analysis of 10 S. uberis strains  as well as their antimicrobial profiles. The authors discovered 10 novel sequences and genes, and "mobile genetic elements previously associated with  antimicrobial resistance".

The manuscript was presented in a well-defined scientific manner. The materials and method section was written with the highest level of scientific rigor in mind. The experiments are replicable and the sample procurement practices were done according to well-defined protocols. 

The authors elaborated on their process for whole genome sequencing, assembly and annotation using standardized methods.  The result section was well defined.

As for the statistical analysis, the authors explained that a  univariate descriptive statistical analysis was carried out by the authors for the experimentations. Although it would have been better if the authors can explain the statistical analysis better, stating their statistical results, p values and other statistical values to establish rigor. 

TM II DNA Library Prep Kit

Author Response

Thank you for your comments and suggestions.

Comments 1: Although it would have been better if the authors can explain the statistical analysis better, stating their statistical results, p values and other statistical values to establish rigor. 

Response 1: The statistical analysis was adjusted (lines 340 - 343).

Reviewer 3 Report

Comments and Suggestions for Authors

The manuscript provides a thorough and insightful overview of the study's findings and their significance for future research and practical applications in Streptococcus uberis and antibiotic resistance. The discussion on genetic diversity, mobile genetic elements, and pathogen adaptability is valuable and relevant. However, the chapter could benefit from further clarification on the practical implications of the findings for infection control, prevention of antibiotic resistance, and the development of improved diagnostic tools for farmers and veterinarians.

Specific comments/recommendations:

One impotant part of the manuscript is antibiotic resistance and genes related to that. Therefore, I think that some information and citations are missing. For example information from these articles:

Myrenas et al., 2025, Veterinary Microbiology, DOI: 10.1016/j.vetmic.2024.110319

Hyeon et al., 2024, Veterinary Research, DOI: 10.1186/s13567-024-01302-0

Zouharova et al., 2023, Antibiotics, DOI: 10.3390/antibiotics12101527

Author Response

Thank you for your comments and suggestions.

Comments 1: However, the conclusion could benefit from further clarification on the practical implications of the findings for infection control, prevention of antibiotic resistance, and the development of improved diagnostic tools for farmers and veterinarians.

Response 1: The conclusion was adjusted (lines 345 - 350).

Comments 2: One important part of the manuscript is antibiotic resistance and genes related to that. Therefore, I think that some information and citations are missing. 

Response 2: The articles suggested were included in the discussion section.

Round 2

Reviewer 1 Report

Comments and Suggestions for Authors

The manuscript has significantly improved, and the author has appropriately addressed my previous concerns. I recommend accepting the manuscript in its current form.